# CHARACTERIZING LINEAR CONVERGENCE IN OPTIMIZATION: POLYAK-ŁOJASIEWICZ INEQUALITY AND WEAK-QUASI-STRONG-CONVEXITY

## ABSTRACT

We give a complete characterization of optimization problems that can be solved by gradient descent with a linear convergence rate. We show that the well-known Polyak-Łojasiewicz inequality is necessary and sufficient for linear convergence with respect to function values to the minimum, while a property that we call "weak-quasi-strong-convexity", or WQSC, is necessary and sufficient for linear convergence with respect to distances of the iterates to an optimum.

## 1 INTRODUCTION AND BACKGROUND

Gradient descent is the most popular algorithm for optimizing a lower bounded and smooth function $f : \mathbb{R}^n \to \mathbb{R}$. It starts from an initial guess and it updates in the direction opposite of the gradient:

$$\tilde{x} = x - \eta \nabla f(x), \tag{1}$$

where $\eta$ is the step size, also called the learning rate.

For many modern applications, gradient descent is overly simplistic. It forms, however, the basis for many practical algorithms like stochastic, distributed or online gradient descent. While favourable convergence guarantees for gradient descent have been long associated with convexity of the objective function $f$, Polyak (1963) showed that a much more general property is sufficient to guarantee linear convergence with respect to the values of $f$ to the minimum $f^*$. The usual result that can be found in the literate is for step size $\eta = \frac{1}{L}$. Below, we present a slightly more general result for any $\eta \leq \frac{2}{L}$.

**Proposition 1** (Polyak, 1963). *Let $f : \mathbb{R}^n \to \mathbb{R}$ be continuously differentiable and bounded below. Consider the optimization problem with minimum $f^*$,*

$$\min_{x \in \mathbb{R}^n} f(x).$$

*Assume also that the gradient of $f$ satisfies for some $L > 0$*

$$\|\nabla f(x) - \nabla f(y)\| \leq L\|x - y\|, \quad \forall x, y \in \mathbb{R}^n \tag{2}$$

*and for some $\mu > 0$*

$$\|\nabla f(x)\|^2 \geq 2\mu(f(x) - f^*), \quad \forall x \in \mathbb{R}^n. \tag{3}$$

*Then, an iterate $\tilde{x}$ of gradient descent starting from $x$ with step size $0 \leq \eta \leq \frac{2}{L}$, satisfies*

$$f(\tilde{x}) - f^* \leq (1 - \eta(2 - \eta L)\mu) (f(x) - f^*).$$

*Proof.* The Lipschitz continuity of the gradient implies (see, e.g., Nesterov (2013), Theorem 2.1.5)

$$f(\tilde{x}) \leq f(x) + \langle \nabla f(x), \tilde{x} - x \rangle + \frac{L}{2}\|\tilde{x} - x\|^2. \tag{4}$$

Since $\tilde{x} - x = -\eta \nabla f(x)$, this gives with equation 3 that

$$f(\tilde{x}) \leq f(x) - \left(\eta - \frac{\eta^2 L}{2}\right) \|\nabla f(x)\|^2 \leq f(x) - \eta(2 - \eta L)\mu(f(x) - f^*).$$

For the last inequality, we take into account that $0 \leq \eta \leq \frac{2}{L}$. Subtracting $f^*$ from both sides gives the desired result. $\qquad\square$

As the condition of Lipschitz continuity of the gradient ($L$-smoothness) is standard and satisfied by most problems in practice, the second condition

$$\|\nabla f(x)\|^2 \geq 2\mu(f(x) - f^*)$$

became known as Polyak–Łojasiewicz (PL) (as at the same time with Boris Polyak, Stanisław Łojasiewicz studied more general versions of this condition). The PL condition with constant $\mu$ is weaker than $\mu$-strong-convexity and does not require the existence of a unique minimizer. It implies however that all critical points of $f$ must be global minima, thus also that all local minima are global.

More recently, another generalization of the notion of strong convexity which relates to convergence with respect to distances, has been studied (see Bu & Mesbahi (2020); Necoara et al. (2019) for similar versions). We define it below:

**Definition 2** (Weak-quasi-strong-convexity (WQSC)). *A function $f \colon \mathbb{R}^n \to \mathbb{R}$ with a convex set of global optima $X^* := \arg\min_{x \in \mathbb{R}^n} f(x)$ is called $(a, \mu)$-weak-quasi-strongly-convex (WQSC), if there exist constants $a, \mu > 0$ such that*

$$f(x) - f^* \leq \frac{1}{a}\langle \nabla f(x), x - x_p \rangle - \frac{\mu}{2}\|x - x_p\|^2, \quad \forall x \in \mathbb{R}^n,$$

*where $x_p$ is the projection of $x$ onto $X^*$.*

**Remark:** In anything that has to do with weak-quasi-strong-convexity, we assume that the set of optima of $f$ is convex. This ensures that the projection $x_p$ of $x$ onto this set is unique. It is not totally clear to us whether this assumption is necessary, but certainly saves from many complications. It is also much more general than assuming that the function $f$ itself is convex (as in Necoara et al. (2019)).

WQSC includes all strongly convex functions, but it also implies PL:

**Proposition 3.** *If $f$ is $(a, \mu)$-WQSC, then it also satisfies the PL condition*

$$\|\nabla f(x)\|^2 \geq 2\mu a^2(f(x) - f^*).$$

*Proof.* The proof is essentially the same with the one of Lemma 3.2 in Bu & Mesbahi (2020). We recall it here, since, in our case, we allow for multiple global optima.

If $f$ is $(a, \mu)$-WQSC, then we have

$$f(x) - f^* \leq \frac{1}{a}\langle \nabla f(x), x - x_p \rangle - \frac{\mu}{2}\|x - x_p\|^2, \quad \forall x \in \mathbb{R}^n,$$

where $x_p$ is the projection of $x$ onto the set of global optima.

We can write

$$\langle \nabla f(x), x - x_p \rangle \leq \frac{\rho}{2}\|\nabla f(x)\|^2 + \frac{1}{2\rho}\|x - x_p\|^2,$$

for all $\rho > 0$.

Combining the two inequalities, we get

$$f(x) - f^* \leq \frac{\rho}{2a}\|\nabla f(x)\|^2 + \frac{1}{2a\rho}\|x - x_p\|^2 - \frac{\mu}{2}\|x - x_p\|^2.$$

Choosing $\rho = \frac{1}{a\mu}$, the two last terms in the right hand side cancel out, and the inequality becomes

$$f(x) - f^* \leq \frac{1}{2a^2\mu}\|\nabla f(x)\|^2,$$

which gives the desired result after a rearrangement.

$\square$

Moreover, WQSC is sufficient for linear convergence of gradient descent with respect to distances of the iterates to the set of optima $X^*$:

**Proposition 4.** *Consider the optimization problem*

$$\min_{x \in \mathbb{R}^n} f(x),$$

*where $f$ is L-smooth and $(a, \mu)$-WQSC. An iterate $\tilde{x}$ of 1 starting from $x$ with step size $0 \leq \eta \leq a/L$ satisfies*

$$\|\tilde{x} - \tilde{x}_p\|^2 \leq (1 - a\mu\eta)\|x - x_p\|^2.$$

*Here, $\tilde{x}_p$ is the projection of $\tilde{x}$ onto $X^* = \arg\min_{x \in \mathbb{R}^n} f(x)$, while $x_p$ is that of $x$.*

*Proof.* The proof is a simple adaptation of Lemma 4.2 in Bu & Mesbahi (2020). The difference is that, in this result, the global optimum is not necessarily unique. We state it here for completeness.

We inspect the quantity $\|\tilde{x} - x_p\|^2$. We have

$$\|\tilde{x} - x_p\|^2 = \|x - \eta\nabla f(x) - x_p\|^2 = \|x - x_p\|^2 - 2\eta\langle\nabla f(x), x - x_p\rangle + \eta^2\|\nabla f(x)\|^2.$$

Notice that since $f$ is L-smooth, we have (again applying [Theorem 2.1.5 in Nesterov (2013)](#))

$$f^* \leq f\left(x - \frac{1}{L}\nabla f(x)\right) \leq f(x) + \left\langle\nabla f(x), -\frac{1}{L}\nabla f(x)\right\rangle + \frac{1}{2L}\|\nabla f(x)\|^2 = f(x) - \frac{1}{2L}\|\nabla f(x)\|^2.$$

By $(a, \mu)$-WQSC of $f$ (Definition 2), we have

$$-\frac{1}{a}\langle\nabla f(x), x - x_p\rangle \leq f^* - f(x) - \frac{\mu}{2}\|x - x_p\|^2$$

and combining with the previous inequality

$$f(x) - f^* \geq \frac{1}{2L}\|\nabla f(x)\|^2,$$

we get

$$-\frac{1}{a}\langle\nabla f(x), x - x_p\rangle \leq -\frac{1}{2L}\|\nabla f(x)\|^2 - \frac{\mu}{2}\|x - x_p\|^2.$$

We now multiply this inequality by $2\eta a$ on both sides:

$$-2\eta\langle\nabla f(x), x - x_p\rangle \leq -\frac{\eta a}{L}\|\nabla f(x)\|^2 - \eta\mu a\|x - x_p\|^2.$$

Substituting in the initial expression above, we get

$$\|\tilde{x} - x_p\|^2 \leq (1 - a\mu\eta)\|x - x_p\|^2 + \left(\eta^2 - \frac{\eta a}{L}\right)\|\nabla f(x)\|^2$$

and since $0 \leq \eta \leq \frac{a}{L}$, we have

$$\|\tilde{x} - x_p\|^2 \leq (1 - a\mu\eta)\|x - x_p\|^2.$$

By noticing that $\|\tilde{x} - \tilde{x}_p\| \leq \|\tilde{x} - x_p\|$ since $\tilde{x}_p$ is the projection of $\tilde{x}$ to the set of optima, we get the desired result. $\square$

**Contribution:** Our main contribution is to go backwards in both the cases of PL and WQSC conditions. That is, we show that PL is a necessary condition for linear convergence of GD for the values of the objective function to the minimum (Theorem 5), while WQSC is necessary for linear convergence of GD with respect to the distances of its iterates to the set of optima $X^*$ (Theorem 9). We believe that our results is a solid step towards understanding tractability in optimization. They also imply many interesting corollaries, of which two are presented in the appendix. The reason that we present such corollaries in the appendix is in order to keep the main text as simple and elementary as possible. The interested reader though is definitely recommended to take a look at them.

## 2 RELATED WORK

While the necessity of the aforementioned convexity-like properties with respect to the convergence guarantees of gradient descent is a natural question, there is not much in the literature dedicated to it. This is surprising to us, as characterizing solvable optimization problems can save future theoreticians and practitioners much unnecessary effort.

The popular paper by Karimi et al. (2016) shows (in Appendix A) that among all the properties proposed in the literature that guarantee linear convergence for gradient descent, the PL condition is the weakest. Such properties include error bounds Luo & Tseng (1993), essential strong convexity Liu et al. (2014), quadratic growth Anitescu (2000) and the, more general, Kurdyka-Łojasiewicz inequality Attouch et al. (2013); Bolte et al. (2010). Various relationships between these notions have been investigated, see for instance Karimi et al. (2016) (Appendix A), Bolte et al. (2017) and Rebjock & Boumal (2024). Our results close a chapter with respect to the search of new relaxations of strong convexity, at least as far as the widespread notion of linear convergence is considered. If one considers different convergence guarantees, there should be perhaps different convexity-like properties associated with them.

The paper closest to the result of Theorem 5 is Abbaszadehpeivasti et al. (2023) (Theorem 5), which shows essentially the same result, but instead of using the standard $L$-smoothness assumption, it uses a lower bounded curvature assumption (equation (3) in that paper). This lower bounded curvature assumption makes the proof almost direct, but it also removes a lot of the generality of the result. One can easily see a duality here: for showing a linear convergence rate starting from a PL inequality one assumes an *upper bound for the curvature* of the function; for showing a PL inequality starting from a linear convergence rate, ones assumes a *lower bound for the curvature* of the function. Our Theorem 5 shows that for passing from a linear convergence rate to a PL condition, $L$-smoothness is all that is needed in the case of gradient descent and unnatural assumptions like equation (3) of Abbaszadehpeivasti et al. (2023) are not required. In Section 3, we give in addition an example of a function that is $L$-smooth, but does not have a lower bounded curvature. Even if a function does have lower bounded curvature, such a lower bound governs the quality of the derived PL condition in Theorem 5 of Abbaszadehpeivasti et al. (2023), that is, if such lower bound is poor, the derived PL inequality is also poor. The same happens with our Theorem 5, but with respect to $L$-smoothness. If $L$ is too large, our derived PL inequality behaves poorly. However, this is to be expected, as a poor smoothness constant would yield a poor convergence result in Proposition 1. Similarly, complete lack of $L$-smoothness ($L = \infty$) would yield to a failure for our Theorem 5, but also for Proposition 1.

The papers closest to the result of Theorem 9 is Alimisis (2024a) and Necoara et al. (2019). Theorem 5 in Alimisis (2024a) shows essentially the same result with our Theorem 9 but assuming that the optimization problem has a unique optimum. It turns out that this assumption can be substituted by the more general one of the set of optima $X^*$ being convex. An interesting class of functions with potentially multiple optima, which indeed form a convex set, is quasi-convex functions. Theorem 5 of Necoara et al. (2019) shows that a convex optimization problem for which GD converges linearly for distances of the iterates to an optimum must satisfy a quadratic growth condition.

Necoara et al. (2019) also present a property they call "quasi-strong-convexity" (Definition 1), which is a slightly stronger version of our property (Definition 2). The latter turns out to be a valuable relaxation of the property presented in Necoara et al. (2019) (and in other works). A first hint on that are the results of Alimisis & Vandereycken (2024b) on the geodesic "weak-strong-convexity" of the symmetric eigenvalue problem. There, the authors show that the symmetric eigenvalue problem is WQSC on the Grassmann manifold in the sense defined in Definition 2. A second hint is given by Alimisis (2024a) and the current work: while both the property of Definition 2 and the quasi-strong-convexity of Necoara et al. (2019) are sufficient to guarantee linear convergence of 1 with respect to distances, only the first can be shown to be also necessary. The WQSC property defined in this work is essentially a generalization of the notion appeared in Bu & Mesbahi (2020) and Alimisis (2024a) to the case of an optimization problem with multiple optima that form a convex set.

It is interesting to note that WQSC can be used to analyze even Nesterov's accelerated gradient descent Bu & Mesbahi (2020); Alimisis et al. (2024c). Similarly, the PL condition can be used to analyse many different algorithms (see for instance Radhakrishnan et al. (2020); Bassily et al. (2018); Li & Li (2018)). Since in this paper we deal only with the case of gradient descent, a natural question is whether some of our results can be extended to other optimization algorithms.

## 3 NECESSITY OF PL CONDITION

We now present our main result, that is, the PL condition is the minimum requirement for having linear convergence of gradient descent with respect to the values of the objective function:

**Theorem 5.** *Let $f\colon \mathbb{R}^n \to \mathbb{R}$ be continuously differentiable, bounded below and $L$-smooth (see equation 2). For $E \subseteq \mathbb{R}^n$ an open and convex set, consider the problem*

$$\min_{x \in E} f(x).$$

*Assume that there exists a step size $\eta$ and constant $c$ such that each iterate $\tilde{x}$ of 1 started from any $x \in E$ satisfies*

$$f(\tilde{x}) - f^* \leq (1-c)(f(x) - f^*).$$

*Then, $f$ satisfies the following PL condition:*

$$\|\nabla f(x)\|^2 \geq \frac{2c}{2\eta + 3\eta^2 L}(f(x) - f^*), \quad \forall x \in E.$$

*Proof.* By $L$-smoothness of $f$, we have

$$\|\nabla f(\tilde{x}) - \nabla f(x)\| \leq L\|\tilde{x} - x\| = \eta L\|\nabla f(x)\|.$$

Since $\|\nabla f(\tilde{x}) - \nabla f(x)\| \geq \|\nabla f(\tilde{x})\| - \|\nabla f(x)\|$, the above gives $\|\nabla f(\tilde{x})\| \leq (1 + \eta L)\|\nabla f(x)\|$.

Again by $L$-smoothness, we have that, for any $x$ and $y$ in $\mathbb{R}^n$, it holds (see equation 4)

$$f(y) \leq f(x) + \langle \nabla f(x), y - x \rangle + \frac{L}{2}\|y - x\|^2.$$

Substituting $y \rightsquigarrow x$ and $x \rightsquigarrow \tilde{x}$, we get

$$f(x) \leq f(\tilde{x}) + \langle \nabla f(\tilde{x}), x - \tilde{x} \rangle + \frac{L}{2}\|x - \tilde{x}\|^2$$

$$= f(\tilde{x}) + \eta\langle \nabla f(\tilde{x}), \nabla f(x) \rangle + \eta^2 \frac{L}{2}\|\nabla f(x)\|^2$$

$$\leq f(\tilde{x}) + \eta\|\nabla f(\tilde{x})\|\|\nabla f(x)\| + \eta^2 \frac{L}{2}\|\nabla f(x)\|^2$$

$$\leq f(\tilde{x}) + \eta(1 + \eta L)\|\nabla f(x)\|^2 + \eta^2 \frac{L}{2}\|\nabla f(x)\|^2$$

$$= f(\tilde{x}) + \left(\eta + \frac{3\eta^2 L}{2}\right)\|\nabla f(x)\|^2.$$

The second inequality is obtained by Cauchy-Schwarz and the third one by our previous bound between $\|\nabla f(\tilde{x})\|$ and $\|\nabla f(x)\|$.

Rearranging, we have

$$f(x) - f(\tilde{x}) \leq \left(\eta + \frac{3\eta^2 L}{2}\right)\|\nabla f(x)\|^2.$$

By the assumption of linear convergence of one iterate of gradient descent, we have

$$f(x) - f^* \leq \frac{1}{c}(f(x) - f(\tilde{x}))$$

and plugging in the previous inequality, we get

$$f(x) - f^* \leq \frac{1}{c}\left(\eta + \frac{3\eta^2 L}{2}\right)\|\nabla f(x)\|^2.$$

This shows that $f$ satisfies a PL condition in $x$ and since the assumption of linear convergence is made for all $x \in E$, $f$ satisfies a PL condition in $E$. Rearranging the last inequality, we get the desired result. $\square$

We now examine whether the previous result is tight in the sense that it gives back the same PL inequality that is used to derive a linear rate. For that, we assume that $c = \eta(2 - \eta L)\mu$ corresponds to the contraction in Proposition 1, and that this rate is attained for all $\eta \leq 2/L$.

**Corollary 6.** *Let $f$ satisfy the hypotheses in Theorem 5. Suppose that gradient descent on $f$ contracts in function value for any $x \in E$ at least as*

$$f(\tilde{x}) - f^* \leq (1 - \eta(2 - \eta L)\mu)(f(x) - f^*), \quad \forall \eta \in (0, 2/L).$$

*Then $f$ is PL on $E$ with constant $\mu$:*

$$\|\nabla f(x)\|^2 \geq 2\mu(f(x) - f^*).$$

*Proof.* By Theorem 5, $f$ satisfies the PL condition

$$\|\nabla f(x)\|^2 \geq \frac{\eta(2 - \eta L)\mu}{\eta + \frac{3\eta^2 L}{2}}(f(x) - f^*) = \frac{2 - \eta L}{1 + \frac{3\eta L}{2}}\mu(f(x) - f^*).$$

Since this result holds for all $\eta$ arbitrarily close to 0, we can take the limit $\eta \longrightarrow 0$. $\qquad\square$

**Remark:** Theorem 5 does not assume that the curvature of the function $f$ is lower bounded by some constant globally, as it is assumed in Abbaszadehpeivasti et al. (2023). There are notable functions, which satisfy our set of assumptions, but not the one of Abbaszadehpeivasti et al. (2023). Consider for instance the function

$$f(x) = (x^2 - x^{1/3}\sin(x))\frac{1}{1 + x^2}.$$

This function is lower bounded with two symmetric optima at around $0.4167$ and $-0.4167$. Its second derivative,

$$f''(x) = \frac{(9x^6 - 22x^4 + 43x^2 + 2)\sin(x) + (30x^5 + 24x^3 - 6x)\cos(x) - 54x^{11/3} + 18x^{5/3}}{9x^{5/3}(x^2 + 1)^3}.$$

is defined on $\mathbb{R}$ except 0. It is upper bounded globally, but it holds $\lim_{x\to 0} f''(x) = -\infty$. Thus, this function does not have a lower bounded curvature globally and does not satisfy the relevant assumption (Equation 3) of Abbaszadehpeivasti et al. (2023).

We now discuss an even more important disadvantage of the result of Abbaszadehpeivasti et al. (2023). The assumed lower bound $\bar{\mu}$ for the curvature (we denote it $\bar{\mu}$ to avoid confusion with the constant $\mu$ used along the rest of our paper) governs the quality of the derived PL condition. This is not clear in Theorem 5 of Abbaszadehpeivasti et al. (2023), as the result is stated for step size $\frac{1}{L}$. We state here the generalization of this result for a general $\eta > 0$, whose proof is essentially the same.

**Proposition 7** (Theorem 5 of Abbaszadehpeivasti et al. (2023) but with general step size)**.** *Consider a lower bounded and $L$-smooth function $f : \mathbb{R}^n \to \mathbb{R}$ with curvature lower bounded by $\bar{\mu}$,*

$$f(y) \geq f(x) + \langle \nabla f(x), y - x \rangle + \frac{\bar{\mu}}{2}\|y - x\|^2, \text{ for all } x, y \in \mathbb{R}^n. \tag{5}$$

If an iterate of 1 satisfies

$$f(\tilde{x}) - f^* \leq (1 - c)(f(x) - f^*)$$

then $f$ satisfies the PL inequality

$$\|\nabla f(x)\|^2 \geq \frac{c}{\eta - \frac{\bar{\mu}}{2}\eta^2}(f(x) - f^*).$$

*Proof.* By equation 5, we have

$$f(\tilde{x}) \geq f(x) + \langle \nabla f(x), \tilde{x} - x \rangle + \frac{\bar{\mu}}{2}\|\tilde{x} - x\|^2 = f(x) - \left(\eta - \frac{\bar{\mu}}{2}\eta^2\right)\|\nabla f(x)\|^2.$$

Thus, we have

$$f(x) - f(\tilde{x}) \leq \left(\eta - \frac{\bar{\mu}}{2}\eta^2\right)\|\nabla f(x)\|^2.$$

By the assumed convergence rate, we have

$$f(x) - f^* \leq \frac{1}{c}(f(x) - f(\tilde{x})) \leq \frac{1}{c}\left(\eta - \frac{\bar{\mu}}{2}\eta^2\right)\|\nabla f(x)\|^2.$$

Rearranging, we get the desired PL inequality. □

We see easily that if the contraction rate $c$ is proportional to $\eta$ or to $\eta(2 - \eta L)$ (as it is in Proposition 1), the PL constant satisfies

$$\frac{c}{\eta - \frac{\bar{\mu}}{2}\eta^2} \leq \mathcal{O}\left(-\frac{1}{\bar{\mu}}\right).$$

Thus, the quality of the derived PL condition worsens considerably for a function that attains negative curvature of large magnitude. In contrast, our result (Theorem 5) features a condition that does not depend on a lower bound for the curvature of $f$ at all.

**Remark:** Notice that the previous result does not use at all the $L$-smoothness assumption, but only the assumption of $\bar{\mu}$ lower bounded curvature. In some sense, it can be seen as the other side of the coin of our Theorem 5. If $L$ is excessively large, our Theorem 5 provides a PL inequality with a bad constant, while if $L = \infty$ (that is, $f$ does not have Lipschitz continuous gradient) Theorem 5 fails. However, $L$-smoothness is the standard assumption used in analyzing gradient descent in the first place (Proposition 1), while a lower bound on the curvature is irrelevant for providing a linear convergence rate for gradient descent to the best of our knowledge. To be more precise, if Theorem 5 fails, so does Proposition 1. Also, if Theorem 5 provides a PL inequality with a very bad constant, then Proposition 1 provides linear convergence with a very bad contraction rate (because the step size must be taken very small).

Given our Theorem 5, we can safely write the following result, completely removing the unnatural lower bounded curvature assumption:

**Theorem 8.** *Consider an $L$-smooth function $f : \mathbb{R}^n \to \mathbb{R}$, continuously differentiable and bounded from below (attaining an optimum $f^*$). Then, gradient descent 1 converges linearly with respect to the values of the function $f$ if and only if it satisfies a PL condition.*

*Proof.* The proof is a simple combination of Proposition 1 and Theorem 5. □

## 4 NECESSITY OF WQSC CONDITION

Our other main result is an analogue to Theorem 5, but now with respect to a contraction for the distance to the set of optima instead of error in function value. This is a generalization of Theorem 5 in Alimisis (2024a) in the case of optimization problems with multiple global optima, after making a few extra mild assumptions. Except that the set of optima is convex, we also assume that the contraction is proportional to the step size $\eta$ and that such contraction happens for all $\eta$ arbitrarily close to 0. These assumptions are mild as they belong in the set of outcomes of Proposition 4.

**Theorem 9.** *Let $f : \mathbb{R}^n \to \mathbb{R}$ be continuously differentiable, bounded below, $L$-smooth (see equation 2) and the set of its global optima $X^*$ is convex. For $E \subseteq \mathbb{R}^n$ an open and convex set, consider the problem*

$$\min_{x \in E} f(x).$$

*Assume that there exist constants $\overline{\eta}$ and $d$ such that each iterate $\tilde{x}$ of 1 started from any $x \in E$ and with any step size $\eta \in (0, \overline{\eta})$ satisfies*

$$\|\tilde{x} - \tilde{x}_p\|^2 \leq (1 - d\eta)\|x - x_p\|^2,$$

*where $x_p$ and $\tilde{x}_p$ are the projections of $x$ and $\tilde{x}$ respectively onto $X^*$. Then, $f$ is $(a, \mu)$-WQSC in $E$ with parameters*

$$a := \frac{d}{2L} \quad , \quad \mu := \frac{L}{2}.$$

*Proof.* Let $x \in E$ and $\tilde{x}$ the result of one iteration of gradient descent 1.

We first rewrite the term $\|\tilde{x} - \tilde{x}_p\|^2$:

$$\|\tilde{x} - \tilde{x}_p\|^2 = \|x - \eta\nabla f(x) - \tilde{x}_p\|^2$$
$$= \|x - \tilde{x}_p\|^2 - 2\eta\langle\nabla f(x), x - \tilde{x}_p\rangle + \eta^2\|\nabla f(x)\|^2.$$

For ease of notation, we set $c := d\eta$. This equality together with the contraction assumption gives

$$\|x - \tilde{x}_p\|^2 - 2\eta\langle\nabla f(x), x - \tilde{x}_p\rangle + \eta^2\|\nabla f(x)\|^2 \leq (1-c)\|x - x_p\|^2 \leq (1-c)\|x - \tilde{x}_p\|^2.$$

The second inequality follows from the fact that $x_p$ is the projection of $x$ onto $X^*$ (as defined in Definition 2). The derived inequality can thus be rewritten as

$$2\eta\langle\nabla f(x), x - \tilde{x}_p\rangle \geq c\|x - \tilde{x}_p\|^2 + \eta^2\|\nabla f(x)\|^2. \tag{6}$$

Next, we use the inequality

$$\langle y, z\rangle \leq \frac{\rho}{2}\|y\|^2 + \frac{1}{2\rho}\|z\|^2$$

that holds for all $y, z \in \mathbb{R}^n$ and any $\rho > 0$ to obtain

$$\frac{\rho}{2}\|\nabla f(x)\|^2 \geq \langle\nabla f(x), x - \tilde{x}_p\rangle - \frac{1}{2\rho}\|x - \tilde{x}_p\|^2.$$

Multiplying both sides by $\frac{2\eta^2}{\rho}$, we get

$$\eta^2\|\nabla f(x)\|^2 \geq \frac{2\eta^2}{\rho}\langle\nabla f(x), x - \tilde{x}_p\rangle - \frac{\eta^2}{\rho^2}\|x - \tilde{x}_p\|^2.$$

Using equation 6, we get

$$2\eta\langle\nabla f(x), x - \tilde{x}_p\rangle \geq c\|x - \tilde{x}_p\|^2 + \frac{2\eta^2}{\rho}\langle\nabla f(x), x - \tilde{x}_p\rangle$$
$$- \frac{\eta^2}{\rho^2}\|x - \tilde{x}_p\|^2,$$

or equivalently

$$\left(2\eta - \frac{2\eta^2}{\rho}\right)\langle\nabla f(x), x - \tilde{x}_p\rangle \geq \left(c - \frac{\eta^2}{\rho^2}\right)\|x - \tilde{x}_p\|^2.$$

Since the last inequality holds for any $\rho > 0$, we choose $\rho := \frac{2\eta}{\sqrt{c}}$ so that is becomes

$$2\eta\left(1 - \frac{\sqrt{c}}{2}\right)\langle\nabla f(x), x - \tilde{x}_p\rangle \geq \frac{3c}{4}\|x - \tilde{x}_p\|^2$$
$$= \frac{c}{4}\|x - \tilde{x}_p\|^2 + \frac{c}{2}\|x - \tilde{x}_p\|^2.$$

By $L$-smoothness of $f$ we have (substitute $y \rightsquigarrow x, x \rightsquigarrow \tilde{x}_p$ in Equation (2.1.6) of Theorem 2.1.5 of Nesterov (2013))

$$\|x - \tilde{x}_p\|^2 \geq \frac{2}{L}(f(x) - f^*),$$

and using that to bound the last term of the previous inequality, we have

$$2\eta\left(1 - \frac{\sqrt{c}}{2}\right)\langle\nabla f(x), x - \tilde{x}_p\rangle \geq \frac{c}{4}\|x - \tilde{x}_p\|^2 + \frac{c}{L}(f(x) - f^*).$$

Rearranging, we get

$$f(x) - f^* \leq 2L\eta\frac{1 - \frac{\sqrt{c}}{2}}{c}\langle\nabla f(x), x - \tilde{x}_p\rangle - \frac{L}{4}\|x - \tilde{x}_p\|^2.$$

Since $c = d\eta$, we substitute and obtain

$$f(x) - f^* \leq 2L\frac{1 - \frac{\sqrt{d\eta}}{2}}{d}\langle\nabla f(x), x - \tilde{x}_p\rangle - \frac{L}{4}\|x - \tilde{x}_p\|^2.$$

Taking the limit $\eta \longrightarrow 0$ in both sides of the inequality, we have that $\tilde{x} \longrightarrow x$ and, since the metric projection onto a convex set is a continuous function, also that $\tilde{x}_p \longrightarrow x_p$. Putting everything together, we have the desired result:

$$f(x) - f^* \leq \frac{2L}{d}\langle\nabla f(x), x - x_p\rangle - \frac{L}{4}\|x - x_p\|^2. \qquad \square$$

**Remark:** Notice that Theorem 9 is not tight, in the sense that if a function is $(a, \mu)$-WQSC with $a = \frac{d}{2L}$ and $\mu = \frac{L}{2}$, then Proposition 4 guarantees a linear rate for the distances of the iterates to $X^*$ with contraction

$$1 - a\mu\eta = 1 - \frac{d}{2L}\frac{L}{2}\eta = 1 - \frac{d\eta}{4}.$$

By a factor $1/4$, this is a slightly worse rate than the one assumed in Theorem 9.

## 5 CONCLUSION

We end with some possible directions for future work.

- What form do our results take if gradient descent is substituted by some other algorithm, possibly gradient-related? For instance, what about stochastic gradient descent and convergence in expectation?

- What form do our results take if the linear convergence guarantees are substituted by some other convergence guarantees? For instance, what would be the connection between weak-quasi-convexity Guminov et al. (2023) and an algebraic convergence rate? We conjecture that the most promising way to phrase such results is through an assumption on the behaviour of some energy functional which describes the optimization problem.

- What is the deeper intuitive meaning of PL and WQSC inequalities that makes them useful for optimization?

- Can we find PL or WQSC models in deep learning? There is already some work on that Soltanolkotabi et al. (2018) (Lemma 7.12), but general understanding is largely missing.

We believe that a thorough exploration of the fundamentals of optimization has a lot to offer in the optimization practice. We hope that our work answers a few meaningful questions in this direction and also creates enough stimulus for future work.

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

## A  A COROLLARY ABOUT OVER-PARAMETRIZED DEEP NEURAL NETWORKS

A prominent direction in deep learning research that attempts to explain phenomena of tractability in training deep neural networks is over-parametrization. Researchers have produced results about all kinds of architectures and all kinds of activation functions, where, if the parameters of the model are sufficiently more than the number of the training data, then such neural networks are easy to train fast. Here we present an example of how Theorem 5 can be used in combination with results of this field to produce interesting new corollaries. We list two of such results in a simplified manner. The reader is referred to the cited papers in order to inspect the exact formulation of the theorems and their proofs.

**Proposition 10** (Du et al. (2019), Theorem 7.1 simplified). *For a convolutional ResNet architecture with smooth activation and sufficiently large width per layer (polynomial in the number of training data) we have that:*

- *There is a set of parameters that achieves $0$ training loss.*

- *Gradient descent from a random initial guess and ran with a sufficiently small step size estimates such set of parameters with linear convergence rate*

$$L(\tilde{\theta}) \leq \left(1 - \frac{\eta\lambda_{min}(K^{(H)})}{2}\right) L(\theta),$$

*where $L$ is the objective function, $\tilde{\theta}$ an iterate of gradient descent starting from $\theta$ and $K^{(H)}$ a symmetric and positive-definite matrix which depends on the neural network architecture. This rate holds with high probability over $\theta$.*

**Proposition 11** (Bassily et al. (2018), Theorem 1). *Consider mini-batch stochastic gradient descent with smooth losses. Suppose that a neural network can achieve $0$ training loss and that the empirical risk function $L$ is PL such that*

$$\|\nabla L(\theta)\|^2 \geq \alpha L(\theta)$$

*for some fixed $\alpha > 0$. Assume also that the global training loss is $\lambda$-smooth and each local loss is $\beta$-smooth. For any mini-batch size $m \in \mathbb{N}$, the mini-batch stochastic gradient descent with constant step size $\eta^*(m) = \frac{\alpha m}{\lambda(\beta+\lambda(m-1))}$ gives the convergence rate*

$$\mathbb{E}[L(\tilde{\theta})] \leq \left(1 - \frac{\alpha\eta^*(m)}{2}\right) L(\theta).$$

Combining these two results with our Theorem 5, we derive the following result:

**Theorem 12.** *A convolutional ResNet architecture satisfying the properties described in Proposition 10 and which has also $\beta$-smooth local loss functions and $\lambda$-smooth global loss can be optimized by mini-batch stochastic gradient descent with batch size $m$ and step size*

$$\eta^*(m) = \frac{\lambda_{\min}(K^{(H)})m}{2\lambda(\beta + \lambda(m-1))}$$

*with the rate*

$$\mathbb{E}[L(\tilde{\theta})] \leq \left(1 - \frac{\lambda_{\min}(K^{(H)})\eta^*(m)}{4}\right) L(\theta),$$

*with high probability over $\theta$.*

*Proof.* By Proposition 10, we have that gradient descent converges with respect to the values of the loss function with contraction $1 - c$ with $c = \frac{\eta\lambda_{\min}(K^{(H)})}{2}$. This holds with high probability over the starting point of the iteration. Assuming also smoothness and combining that with our Theorem 5, we have that $L$ satisfies the following PL condition

$$\|\nabla L(\theta)\|^2 \geq \frac{c}{\eta + \frac{3\eta^2\lambda}{2}} L(\theta) = \frac{\frac{\eta\lambda_{\min}(K^{(H)})}{2}}{\eta + \frac{3\eta^2\lambda}{2}} L(\theta) = \frac{\lambda_{\min}(K^{(H)})}{2 + 3\eta\lambda} L(\theta)$$

with high probability over $\theta$.

As this result holds for all $\eta$ arbitrarily close to $0$, we can take the limit $\eta \longrightarrow 0$ and we have

$$\|\nabla L(\theta)\|^2 \geq \frac{\lambda_{min}(K^{(H)})}{2} L(\theta).$$

Now we can use Proposition 11 with $\alpha = \frac{\lambda_{\min}(K^{(H)})}{2}$ and derive the desired result. $\square$

Thus, despite Du et al. (2019) treat only the case of gradient descent, we see that we can pass to the case of stochastic gradient descent easily using Theorem 5, assuming that the loss function has Lipschitz continuous gradient. We do not check the latter, but we conjecture that it is true, as it is a quite standard assumption.

# B   A FUNDAMENTAL COROLLARY BETWEEN THE TWO TYPES OF CONVERGENCE

We present another interesting corollary: since WQSC is equivalent with linear convergence of gradient descent for distances, PL is equivalent with linear convergence of gradient descent for function values and is stronger than PL, we can pass from a linear rate with respect to distances to a linear rate with respect function values.

**Corollary 13.** *Consider a function $f : \mathbb{R}^n \to \mathbb{R}$ and the problem*

$$\min_{x \in E} f(x),$$

*where $E \subseteq \mathbb{R}^n$ open and convex. If $f$ is L-smooth with a convex set of global optima and an iterate $\tilde{x}$ of 1 starting from any $x \in E$ with any step size $\eta \in (0, 2/L)$ satisfies*

$$\|\tilde{x} - \tilde{x}_p\|^2 \leq (1 - d\eta)\|x - x_p\|^2,$$

*for a constant $d \in (0, 1/\eta)$, then an iterate $\bar{x}$ of 1 starting from $x$ with step size $\bar{\eta} \in (0, 2/L)$ satisfies*

$$f(\bar{x}) - f^* \leq \left(1 - \left(\frac{2}{L} - \bar{\eta}\right)\frac{d^2\bar{\eta}}{8}\right)(f(x) - f^*).$$

*Proof.* Since an iterate of gradient descent contracts with respect to the distance from $X^*$, Theorem 9 implies that $f$ is $(a, \mu)$-WQSC, with $a = \frac{d}{2L}$ and $\mu = \frac{L}{2}$. By Proposition 3, we have that $f$ satisfies the PL condition

$$\|\nabla f(x)\|^2 \geq 2\mu a^2(f(x) - f^*) = 2\frac{L}{2}\frac{d^2}{4L^2}(f(x) - f^*) = \frac{d^2}{4L}(f(x) - f^*).$$

By Proposition 1, we have that $\bar{x} = x - \bar{\eta}\nabla f(x)$ satisfies

$$f(\bar{x}) - f^* \leq \left(1 - \bar{\eta}(2 - \bar{\eta}L)\frac{d^2}{8L}\right)(f(x) - f^*) = \left(1 - \left(\frac{2}{L} - \bar{\eta}\right)\frac{d^2\bar{\eta}}{8}\right)(f(x) - f^*).$$

$\square$

**Remark:**   In Corollary 13 we pass from linear convergence with respect to distances to linear convergence with respect to function values. We lose something from the sharpness of the contraction though: if $\bar{\eta} = \frac{1}{L}$ and $\tilde{x} \equiv \bar{x} = x - \bar{\eta}\nabla f(x)$, then starting from a rate

$$\|\tilde{x} - \tilde{x}_p\|^2 \leq (1 - d\bar{\eta})\|x - x_p\|^2$$

yields to a rate

$$f(\bar{x}) - f^* \leq \left(1 - \frac{d^2\bar{\eta}^2}{8}\right)(f(x) - f^*).$$

As $d\bar{\eta} < 1$, the latter rate is slower.

Even in the case that

