# OpenReview forum: "Characterizing linear convergence in optimization: Polyak-Łojasiewicz inequality and weak-quasi-strong-convexity"
_ICLR.cc/2025/Conference — ICLR 2025 Conference Withdrawn Submission_

### Official Review · Reviewer_1JSR · 2024-10-24

**Soundness:** 4
**Presentation:** 2
**Contribution:** 2
**Rating:** 6
**Confidence:** 4

**Summary:**

The paper studies the linear convergence of gradient descent when applied to minimizing a smooth but possibly nonconvex function. The main contribution is two theoretical results characterizing the convergence rate of gradient descent under the PL inequality or Weak-Quasi-Strong-Convexity (WQSC) inequality.

The first result shows that the PL inequality is both necessary and sufficient for linear convergence of gradient descent, with convergence here meaning in terms of the function values f(x_k) - f*. Of course the sufficiency of PL inequality for this result is well-known but the reverse implication, that linear convergence implies the PL inequality, seems novel.

The second result shows something similar for functions satisfying the (WQSC) property and linear convergence with respect to the distance of the iterates to the set of solutions.

In this way, the authors seem to have solved the search for a "weak PL" condition that will ensure linear convergence - the PL inequality is necessary.

Some applications to neural networks are considered but no numerical investigations are done.

**Strengths:**

- The arguments appear to be correct and the results are stated clearly and precisely.

- The results are novel in the sense that previous results of this nature considered (slightly) narrower function classes. The characterization of linear convergence in terms of PL inequality is significant.

**Weaknesses:**

- The results in this paper are, on the whole, incremental in nature. They mostly extend known results to the case where the set of optima is a convex set rather than assuming that it's a singleton. While it's no doubt an important distinction, the analysis involved is hardly changed.

- The paper is purely theoretical, without any investigations into whether these results hold or give insight in practical scenarios. Although some effort is made in the appendix to discuss neural networks that may satisfy the hypotheses of the results here, they are under assumptions like Lipschitz-continuity of the gradient which is actually not typically satisfied by neural networks.

- The related work is underdeveloped, with no mention of the Kurdyka-Łojasiewicz inequality or the associated results about error bounds and convergence rates. I think there is certainly a connection here that is at least worth referencing, if not developing further.

**Questions:**

- To your knowledge, is the work by Abbaszadehpeivasti et al. the only other result that shows necessity of PL for linear convergence? I was surprised that this has not been examined before given the large amount of literature on the PL (and KL) conditions.

- Are there notable functions covered by your relaxed assumptions that are not covered by the weak convexity assumptions in  Abbaszadehpeivasti et al. ? Their curvature lower bound can have a negative constant so it covers a pretty broad class of functions, c.f. Drusvyatskiy and Paquette 2019, if I am not mistaken.

- Similar question to the above - are there actual modern deep learning problems whose set of minimizers is convex that are not toy problems? It seems a bit exaggerated to consider the work of Alimisis 2024 in line 182 as having unrealistic assumptions for assuming that the minimizer is unique. While it is indeed unrealistic, from the point of view of realism/modern deep learning, I don't see why it's any more unrealistic than assuming that the set of minimizers is convex - as far as I know this is simply not the case in modern deep learning. Furthermore, from the point of view of theory, the difference between having a singleton minimizer or a convex set is small - you have many of the same inequalities just by taking the projection onto the set of minimizers, as is done in the current submission.

*References*

Efficiency of minimizing compositions of convex functions and smooth maps
Drusvyatskiy and Paquette 2019

---

> ### Author Response · Authors · 2024-11-18
>
> Thank you for delving deep into our paper, we really appreciate it!
>
> ### Incremental results:
> We consider Theorem 5 as our main result (not Theorem 9, previously Theorem 7). In any case, we believe that both have publication merits for reason that we highlight in our general response and in our newly added discussion in lines 178-193, 196-198 and 288-355. We would like to (rhetorically) ask the reviewer: if our results are incremental, wouldn't Abbaszadehpeivasti et al. see that their (quite non-standard) assumption of lower bounded curvature is not needed?
>
> ### Practicality:
> Indeed our paper is theoretical, but we believe that the results are of vast importance for optimization in general. They kind of close a chapter on what optimization problems can be solved fast via gradient descent. $L$-smoothness is the most standard assumption in optimization and it is satisfied by many real-world problems. Most importantly, it is the standard assumption used in the analysis of gradient descent (Proposition 1). The reviewer is correct that in deep learning it is hard to be the case, but our deep learning example should not be perceived as a main result, rather as a potential application of this theory. If the reviewer believes that it lacks rigor, we can remove it.
>
> ### Related work:
> Your comment about the KL inequality and error bounds is fair and we cited a few works in this topic in lines 172-175.
>
> ### Previous works on necessity of PL for linear convergence:
>  Yes, to the best of our knowledge Abbaszadehpeivasti et al. is the only paper that poses the question of necessity of PL condition for linear convergence of gradient descent. Another paper that shows a relevant result is Karimi et al. (2016). There, it is shown that among all properties that people have come up to show linear convergence of gradient descent, PL is the weakest. We spent quite some time searching over the literature and nothing else came up. Also, nothing else is cited in Abbaszadehpeivasti et al., which is a paper of 2023, thus we are confident that no other works of this nature exist.  This is surprising also to us.
>
> ### Notable functions covered by our relaxed assumptions, but not covered by Abbaszadehpeivasti et al.:
>  Indeed the lower bounded curvature assumption of Abbaszadehpeivasti et al. is not that bad, there are
> many useful function classes that it happens to be the case, one of which is presented indeed in Drusvyatskiy and Paquette (2019). For examples of functions that do not satisfy this assumption, but they are still lower bounded and $L$-smooth, please refer to our general response and in 288-302 of our revised paper. Please also note the unfavorable (and unnatural) dependence of the derived PL condition to the lower curvature bound as discussed in our general response and in lines 302-355 of our revised paper.
>
> ### Actual modern deep learning problems whose set of minimizers is convex:
> We are not aware of such deep learning models, but our focus is not on deep learning. Our phrase "This assumption is unrealistic in the case of modern
> over-parametrized machine/deep learning models, where optima usually form a continuum" (lines 181-183 in the old version) is misleading and we removed it, as, indeed, from a deep learning perspective both assumptions are equally unrealistic. As mentioned in our general response, an interesting class of functions that have convex set of optima is quasi-convex functions. We added that in the place of the old phrase (lines 196-198 in the new version).

---

### Official Review · Reviewer_JpVE · 2024-10-30

**Soundness:** 1
**Presentation:** 2
**Contribution:** 1
**Rating:** 3
**Confidence:** 4

**Summary:**

This paper characterizes the necessary and sufficient condition for the GD to have a linear convergence for smooth (possibly non-convex) functions with respect to the function value and the distances between the iterate and the optimum (that is a projection of the iterate to the solution set). The PL and WQSC conditions are known to provide a linear convergence for GD, and this paper shows that they are also necessary conditions.

**Strengths:**

See Weaknesses.

**Weaknesses:**

Theorems 5 and 7 are the main contributions of this paper, but they appear to be either equivalent or nearly identical to the closest results in Abbaszadehpeivasti et al. (2023) and Alimisis (2024a) mentioned by the authors.

- Theorem 5: The authors mention that Abbaszadehpeivasti et al. (2023) assumes extra assumption (see line 177), but I found that this is just a corollary of the Lipschitz gradient assumption. Theorem 5 in Abbaszadehpeivasti et al. (2023) therefore essentially assumes the same condition as Theorem 5 in this paper, so I should say that there is nothing new in Theorem 5.

- Theorem 7: The only difference between Theorem 7 of this paper and Theorem 5 in Alimisis (2024a) is the assumption on solution set; the former assumes that the solution set is convex, while the latter assumes that the solution set is a singleton set. Although I agree with the authors that their assumption is indeed weaker, I do not see this as a significant advancement, especially given that lines 346-347 were only required to address a non-singleton convex solution set.

Apart from the above, I do not find this paper's contributions sufficient for publication in ICLR. I recommend that the authors address the questions raised in the Conclusion to better motivate this work for the machine learning community.

**Questions:**

See Weaknesses.

---

> ### Author Response · Authors · 2024-11-18
>
> Thank you for your review!
>
> ### Theorem 5:
> We would like to kindly point out that the reviewers' argument that the lower bounded curvature assumption is a corollary of the Lipschitz continuity of the gradient (i.e. of an upper bounded curvature assumption) is incorrect. Please refer to our general response for functions that have upper bounded curvature, but not lower bounded one. Our new Theorem 8 (i.e. the merge of Proposition 1 and Theorem 5) definitely features a weaker set of assumptions compared to Theorem 5 of Abbaszadehpeivasti et al. (2023).
>
> ### Theorem 7 (Theorem 9 in the new version):
>  Please refer to our general response for reasons that our result is sufficiently more general compared to the results of Alimisis (2024(a)). We do believe that it has publication merit, but that is of course a subjective argument.
>
> ### Soundness and Presentation:
> The reviewer has indicated the soundness of our paper as poor. Would it be possible to indicate which part of our paper is incorrect? Also, they have given a low score for our presentation. Would it be possible to indicate how our presentation could be improved?

---

### Official Review · Reviewer_TY5U · 2024-11-02

**Soundness:** 2
**Presentation:** 2
**Contribution:** 2
**Rating:** 3
**Confidence:** 4

**Summary:**

The paper presents the necessary and sufficient conditions for the linear convergence of the gradient descent (GD) method. Specifically, the authors prove that the Polyak-Łojasiewicz (PL) condition is a necessary and sufficient condition for the function values to converge linearly to the minimum. In contrast, the weak-quasi-strong-convexity (WQSC) condition is a necessary and sufficient condition for the sequence of iterates to converge to the minimizer.

**Strengths:**

The authors clarify the necessary and sufficient conditions for the linear convergence of GD, including both the convergence of function values and the convergence of the sequence of iterates.

**Weaknesses:**

1. This paper lacks a clear comparison with existing work. Additionally, I am uncertain whether the authors are the first to suggest that the Polyak-Lojasiewicz (PL) condition is necessary for the linear convergence of function values in gradient descent.
2. According to Definition 2, there is only one global solution. If there were two distinct global solutions, the inequality in Definition 2 would not hold. Therefore, the WQSC condition proposed by the authors is essentially equivalent to the weak-strongly convex condition in (Alimisis (2024a) ). Am I correct in this assessment? If so, it should be noted that linear convergence under the weak-strong convexity property has already been established.

**Questions:**

Line 42: Should the stepsize $\eta$ be in the range $(0,2/L)$ or $[0,2/L]$?
Line 46: The authors should reference original or classic sources. Additionally, they cite too many papers from arXiv, raising concerns about the reliability of the results in these papers.
Lines 269 and 346: Should we consider the step length $\eta$ as a constant, or as a variable that can converge to 0? I believe these two cases should not be conflated. Therefore, I suspect there may be an issue with the relevant proof.

---

> ### Author Response · Authors · 2024-11-18
>
> Thank you for your review!
>
> ### "This paper lacks a clear comparison with existing work. Additionally, I am uncertain whether the authors are the first to suggest that the Polyak-Lojasiewicz (PL) condition is necessary for the linear convergence of function values in gradient descent":
> We have done a literature review to the best of our knowledge. We were able to find only some partial results on the topic, but if you have specific suggestions of related work that we missed, we would be of course happy to add it. There was indeed some lack of clarity in the compasiron with the results of Abbaszadehpeivasti et al.(2023), which is now fixed (see our general response and our additions in blue in lines 178-193 and 288-355).
>
> ### "According to Definition 2, there is only one global solution. If there were two distinct global solutions, the inequality in Definition 2 would not hold. Therefore, the WQSC condition proposed by the authors is essentially equivalent to the weak-strongly convex condition in (Alimisis (2024a) )":
> No, this is not correct, Definition 2 does not imply that the set of optima must be a singleton. Please refer to our general response for papers that define WQSC in the way of Definition 2. You can also consider the following example:
> $$ f(x) =
>       (x^2-1)^2, \hspace{1mm} \text{when} \hspace{1mm}  x<-1 \hspace{1mm} \text{or} \hspace{1mm} x>1 \hspace{1mm} \text{and} \hspace{1mm}
>       f(x)=0, \hspace{1mm} \text{when} \hspace{1mm} -1 \leq x \leq 1
> $$
> This function is lower bounded and differentiable, while its set of optima is the whole interval $[-1,1]$. Since it is a quadratic for $x>1$ and for $x<-1$, it is WQSC everywhere in the sense of Definition 2 (but does not have a unique optimum). When $x>1$, the projection onto the set of optima is $x_p=1$, while when $x<-1$, the projection onto the set of optima is $x_p=-1$.
>
> ### line 42:
>  There is nothing wrong with the step size being allowed to become $0$ or $2/L$, but in this case the algorithm does not do any progress, as $1-\eta(2-\eta L) \mu =0$.
>
> ### line 46:
> We referenced the survey by Zhou, since it has a nice exposition of all implications of smoothness and strong convexity. Your comment is fair though and we changed the reference to one from a textbook by Yurii Nesterov.
>
> ### "too many papers from arxiv":
> We cited in total six arxiv preprints (four in the revised version), none of which is central for the current work. Only the preprint by Bu and Mesbahi (2020) is used in our preliminary results, and, even then, we analyze all the needed proofs again in detail. The papers Abbaszadehpeivasti et al.(2023), Alimisis(2024(a)) and Necoara et al.(2019) (which are central in this work) are all published.
>
> ### limit $\eta \longrightarrow 0$:
> In both cases (lines 284 and 428 in our new version), we have previously established an inequality that holds for any fixed $\eta>0$. Thus, it is perfectly fine to consider what happens in the case that $\eta>0$ becomes arbitrarily small, by taking the limit in both sides of these inequalities. Since the inequalities of lines 282 and 426 (in the new version) hold for all $\eta>0$, they will be preserved also when taking the limit $\eta \longrightarrow 0$ and there is no issue in the proofs.

---

> > ### Comment · Reviewer_n8TC · 2024-11-22
> > **limit $\eta \to 0$**
> >
> > Sorry, but I disagree with your last comment. And now I've looked through the proof of Corollary 6 again, and I am also not sure that it is correct. From my point of view, you fix some $\eta \in (0, 2/L)$ in the beginning of the Corollary, and at the end of the proof for some reason you make $\eta$ (which is fixed) go to zero. I don't get it. As far as I understand, If you claim that $\eta$ is fixed, you should say that this proof works only if $\eta \to 0$, and in case when just $\eta \in (0, 2/L)$, you have the inequality on the line 283.
> >
> > You can consider, for example, $L = 0.1$, and $\eta = 2/L - 1 = 19$. Obviously, we cannot say that $\eta \to 0$. Thus, the Corollary is incorrect, since it does not work $\forall \eta \in (0, 2/L)$.
> >
> > On the other hand, I don't understand why do you need this corollary at all. Because if you take $\mu \equiv 2c / (2 \eta + 3 \eta^2 L)$, for $c, \eta$ fixed you get the PL-condition.

---

### Official Review · Reviewer_n8TC · 2024-11-03

**Soundness:** 4
**Presentation:** 4
**Contribution:** 3
**Rating:** 8
**Confidence:** 3

**Summary:**

In this paper authors study the linear convergence conditions for gradient descent algorithm. They prove that well-known PL-condition is not just sufficient, but also necessary for gradient descent to have linear convergence in terms of functional residual. Additionally, authors introduce a new class of problems, that they call weak-quasi-strongly-convex. Authors claim, that on this class of problems gradient descent has linear convergence rate in terms of distance to the solution necessary and sufficiently.

**Strengths:**

The authors introduce two novel theoretical results. Firstly, they show that PL-condition is not just sufficient, but also necessary for linear convergence of gradient descent on smooth function. Secondly, the authors introduce the definition of weak-quasi-strongly-convex (WQSC) function. Moreover, the authors prove that the WQSC condition is necessary and sufficient for gradient descent to converge linearly in terms of distance to the solution. The WQSC condition is less strict than strong convexity, which broadens the class of problems, where we can obtain linear convergence with respect to distance to the solution.
Overall, the result seems significant for the community while being very simple. In theory, it allows researchers to connect theoretical results for linear convergent methods and problems that satisfy the PL condition (see Theorem 10 in Appendix). In practice, the WQSC condition, while being a relaxation of the strong convexity condition, broadens the spectrum of problems, where GD has linear convergence in terms of distance to the solution.

**Weaknesses:**

Lack of experiments. It would be beneficial to provide some experiments on WQSC problem, which show linear convergence of GD on this problem.

**Questions:**

When you mention an equation, please, change \ref to \eqref, since it is a bit confusing.

---

> ### Author Response · Authors · 2024-11-18
>
> Thank you for your review and the positive feedback!
>
> ### Experiments:
> We are not sure what experiments would fit our paper. One usually runs experiments when there are multiple algorithms to be compared for the same problem.  In our case, we only have gradient descent and some theoretical results around it. We could of course implement gradient descent for a WQSC problem and show that Proposition 4 is also numerically true, but we do not see how this would improve our work.
>
> ### ref to eqref:
>  We do use eqref to refer to previous equations, except when we refer to gradient descent (Equation (1)). We think it is better that way, since gradient descent is an algorithm, not exactly an equation.

---

### Author Response · Authors · 2024-11-18
**General response to all reviewers / part 2**

### Added value in a technical level:
We do not claim that our results are groundbreaking in terms of technicality. We prefer though to see their simplicity as an advantage rather than as a disadvantage, as they can be understood by a very wide audience, broadly interested in optimization.

#### Theorem 5:
Abbaszadehpeivasti et al.(2023) show the necessity of PL in their Theorem 5 under the assumption of lower bounded curvature (only), making the proof almost trivial (see also our new Proposition 7 for a slightly more general version). This is because, the lower bound for the curvature (in combination with the choice $1/L$ for the step size) provides the important inequality $$ f(x)-f(\tilde x) \leq \frac{2L - \mu}{2 L^2} \| \nabla f(x) \|^2, $$
where $\tilde x$ is an iterate of gradient descent starting from $x$ with step $1/L$.

We noticed that in the case of gradient descent, deriving such a bound can be obtained only using $L$-smoothness and the specific structure of the algorithm (lines 233-258 in the new version of our paper). This by no means implies that, in general, an upper bound for the curvature is the same as a lower bound for the curvature. For twice differentiable functions, the first corresponds to an upper bound for the eigenvalues of the Hessian, while the latter to a lower bound for them. Recall also the examples in the previous section.

#### Theorem 9 (previously 7):
The main technical backbone of Theorem 9 (previously Theorem 7) has been developed in Alimisis (2024(a)). The technical contribution of our work is a neat way to extend its result in the case of a problem with multiple optima, by making a more restrictive (but totally natural) assumption (line 373 in the new version) on the contraction rate of gradient descent (must be proportional to the step size) and work things in the limit of arbitrarily small step sizes (lines 424-431 in the new version). We believe that such an argument is not trivial.

To conclude, we believe that sometimes it is easy to label a result as incremental, when it is seen written down clearly. While our community certainly suffers from a high supply of incremental results, we are confident that our results are quite important and novel and we give plenty of evidence in our previous discussion. We admit though that certain aspects of these results were made clearer to us after receiving your reviews, and we thank you for that.

---

### Author Response · Authors · 2024-11-18
**General response to all reviewers / part 1**

We thank the reviewers for their time and energy. Their efforts mean a great service to us, as they encouraged us to look our results and related ones in greater detail and make valuable conclusions. We address here some topics of criticism that seem to underline many of your concerns and we also respond to each one of you individually. We have also revised our paper, with changes highlighted in blue.

### Presentation:
We have made an honest effort to discuss related work to the best of our knowledge and we think that the comparison with our work reflects the best of our understanding. We have added some related work about the Kurdyka-Łojasiewicz inequality and its relationship to error bounds as suggested by reviewer 1JSR (lines 172-175 in the new version). It is surprising also to us that such a natural question on necessity of weak convexity notions has not received more attention in the literature. We kindly believe that reviewers' uncertainty over previous work should result in lower confidence score and not in lower presentation score for our paper.

### Added value in a high level:
#### Theorem 5:

An instance of our Theorem 5 can be found in Theorem 5 of Abbaszadehpeivasti et al.(2023), under the assumption of bounded curvature from both above ($L$-smoothness) and below (even by a negative $\mu$). After a closer inspection, we figured out that in Theorem 5 of Abbaszadehpeivasti et al.(2023) only the lower bounded curvature assumption is used to show the necessity part of the result, while $L$-smoothness is used only for the sufficiency part of the result. Their result confused as because it is stated as an "if and only if" statement. We have added a slightly more general version of the necessity part of Theorem 5 of Abbaszadehpeivasti et al. in our revised version of the paper as Proposition 7.
The assumption of lower bounded curvature is in general not standard, in the sense that such assumption cannot be used in the sufficiency part of the result (i.e. to guarantee a linear convergence rate starting from a PL condition as in Proposition 1). In contrast, our result (Theorem 5) derives the necessity of PL condition assuming $L$-smoothness and utilizing the very structure of the gradient descent algorithm. We have added an "if and only if" statement in our revised paper as Theorem 8, where the lower bounded curvature assumption is completely absent.
There are notable functions that are lower bounded, $L$-smooth, but do not have a lower bounded curvature. Here are some examples of such functions $f: \mathbb{R} \rightarrow \mathbb{R}$. This means that they fit the assumptions of our results (Theorems 5 and 8), but not the assumptions of Abbaszadehpeivasti et al.:
>- $f(x)= (x^2-x^{1/3}\sin(x))\frac{1}{1+x^2} $
>- $f(x) =(x^4 - x^{1/5} \sin(x)) e^{-x^2} $
>- $f(x) = x^2 - x^{1/7} \left(x-\frac{x^3}{6} \right) e^{-x^2}$


These functions are all lower bounded, differentiable everywhere, and twice differentiable everywhere except at $x=0$. $f''(x)$ is always upper bounded, thus the curvature is always upper bounded by some $L>0$. Close to $0$ though, it holds $\lim_{x \rightarrow 0} f''(x) = -\infty$. This means that the curvature at $0$ is $-\infty$. We have included the first function as an example in the new version of our paper (lines 288-302).

Even in the case that the curvature does not become $-\infty$ but it is still negative with a very large magnitude, the result by Abbaszadehpeivasti et al. suffers, as the constant with which their derived PL inequality holds is of the order $\mathcal{O}(-1/\bar \mu)$ (line 333 in the new version), where $\bar \mu$ is a lower bound for the curvature.

An overview of the previous discussion can also be found in the new version of our paper in lines 178-193.

#### Theorem 9 (previously Theorem 7):

Regarding Theorem 9 (Theorem 7 in the old version), the added value compared to the result of Alimisis (2024(a)) is that our result holds in the case of multiple optima which form a convex set (in contrast to a singleton).

>- The first reason that we took the time to work in the case of multiple global optima is that the usual definition of WQSC (or whatever various authors choose to name it) involves the possibility of multiple optima and the projection to their set, see Necoara et al. (2019) (Definition 1) and Karimi et al. (2016) (end of page 11). In all these works, one needs to assume that the set of optima is convex, in order to guarantee the uniqueness of the projection.
>- The second reason is that allowing multiple optima, even forming a convex set, is strictly more general than allowing a unique optimum. A prominent class of functions that have convex set of optima is the class of quasi-convex functions (all level sets of a quasi-convex function are convex, thus also the set of global optima). It is our bad that this example was missing from the original paper and we added it in the revised version (lines 196-198).

---

> ### Comment · Reviewer_1JSR · 2024-11-22
>
> I think there is some confusion because of how the word "curvature" was used colloquially in my review. A weakly convex function does need a second derivative to exist or be bounded. The definition of a weakly convex function f is that there exists some r > 0 so that f + rg is convex where g is the squared euclidean norm. Indeed every continuously differentiable function from Rn to R with Lipschitz-continuous gradient is automatically weakly convex, see "Stochastic model-based minimization of weakly convex functions" by Damek Davis, Dmitriy Drusvyatskiy, page 3 it is written, "The class of weakly convex functions, first introduced in English in [50], is broad. It
> includes all convex functions and smooth functions with Lipschitz continuous gradient".
>
> With this in mind, it seems that the results in this submission are indeed already implied by the results of Abbaszadehpeivasti et al.

---

### Note · Authors · 2024-11-23

**Comment:**

After processing the new feedback from the reviewers, we realised that there is indeed an issue with the originality of our Theorem 5. Indeed, Lipschitz continuity of the gradient provides both an upper and lower bound for the curvature of the function. We got confused, because in the convergence analysis of GD via the PL condition only an upper bound for the curvature is used, and we tried to do the same for the inverse result. We did use though the general Lipschitz continuity of the gradient at the very beginning of the proof of Theorem 5, thus both our result and Abbaszadehpeivasti et al. (2023) are similar in generality. Best is to retract the paper and try at least to remove the general Lipschitz continuity of the gradient assumption (i.e. to work only assuming an upper bound for the curvature of the function).

We would like to warmly thank the reviewers, especially 1JSR for pointing out clearly the aforementioned issue. Their diligence is of great value for us.

**Withdrawal Confirmation:**

I have read and agree with the venue's withdrawal policy on behalf of myself and my co-authors.